# Are long telomeres better than short? Relative contributions of genetically predicted telomere length to neoplastic and non-neoplastic disease risk and population health burden

Ekaterina Protsenko[1], David Rehkopf[2], Aric A. Prather[3], Elissa Epel[3], Jue Lin[4]*

**1** UCSF School of Medicine, San Francisco, CA, United States of America, **2** Stanford Department of Primary Care and Population Health, Stanford, CA, United States of America, **3** UCSF Department of Psychiatry, San Francisco, CA, United States of America, **4** UCSF Department of Biochemistry and Biophysics, San Francisco, CA, United States of America

* jue.lin@ucsf.edu

**Data Availability Statement:** All relevant data are within the manuscript and its Supporting Information files.

## Abstract

### Background

Mendelian Randomization (MR) studies exploiting single nucleotide polymorphisms (SNPs) predictive of leukocyte telomere length (LTL) have suggested that shorter genetically determined telomere length (gTL) is associated with increased risks of degenerative diseases, including cardiovascular and Alzheimer's diseases, while longer gTL is associated with increased cancer risks. These varying directions of disease risk have long begged the question: when it comes to telomeres, is it better to be long or short? We propose to operationalize and answer this question by considering the relative impact of long gTL vs. short gTL on disease incidence and burden in a population.

### Methods and findings

We used odds ratios (OR) of disease associated with gTL from a recently published MR meta-analysis to approximate the relative contributions of gTL to the incidence and burden of neoplastic and non-neoplastic disease in a European population. We obtained incidence data of the 9 cancers associated with long gTL and 4 non-neoplastic diseases associated with short gTL from the Institute of Health Metrics (IHME). Incidence rates of individual cancers from SEER, a database of United States cancer records, were used to weight the ORs in order to align with the available IHME data. These data were used to estimate the excess incidences due to long vs. short gTL, expressed as per 100,000 persons per standard deviation (SD) change in gTL. To estimate the population disease burden, we used the Disability Adjusted Life Years (DALY) metric from the IHME, a measure of overall disease burden that accounts for both mortality and morbidity, and similarly calculated the excess DALY associated with long vs. short gTL.

**Funding:** The authors received no specific funding for this work.

**Competing interests:** All authors declare no competing interests for this work.

## Results

Our analysis shows that, despite the markedly larger ORs of neoplastic disease, the large incidence of degenerative diseases causes the excess incidence attributable to gTL to balance that of neoplastic diseases. Long gTL is associated with an excess incidence of 94.04 cases/100,000 persons/SD (45.49–168.84, 95%CI) from the 9 cancer, while short gTL is associated with an excess incidence of 121.49 cases/100,000 persons/SD (48.40–228.58, 95%CI) from the 4 non-neoplastic diseases. When considering disease burden using the DALY metric, long gTL is associated with an excess 1255.25 DALYs/100,000 persons/SD (662.71–2163.83, 95%CI) due to the 9 cancers, while short gTL is associated with an excess 1007.75 DALYs/100,000 persons/SD (411.63–1847.34, 95%CI) due to 4 non-neoplastic diseases.

## Conclusions

Our results show that genetically determined long and short telomere length are associated with disease risk and burden of approximately equal magnitude. These results provide quantitative estimates of the relative impact of genetically-predicted short vs. long TL in a human population, and provide evidence in support of the cancer-aging paradox, wherein human telomere length is balanced by opposing evolutionary forces acting to minimize both neoplastic and non-neoplastic diseases. Importantly, our results indicate that odds ratios alone can be misleading in different clinical scenarios, and disease risk should be assessed from both an individual and population level in order to draw appropriate conclusions about the risk factor's role in human health.

## Introduction

Telomeres are the protective ends of chromosomes consisting of a repeating DNA sequence, which function to preserve genome stability by buffering against the progressive loss of terminal DNA during cell division and other forms of cellular damage [1]. Telomeres shorten as a normal process of human aging, but individuals vary widely in their rate of attrition and in their measured telomere length at any given point in time. Accordingly, measures of telomere length (TL) and rate of attrition have been proposed as biomarkers for risks of age-related diseases [2].

Phenotypically measured telomere length (mTL) is the cumulative result of both genetic and non-genetic contributions [2]. The significant role of inheritance was demonstrated by early reports from twin and sibling studies, which estimated the heritability of telomere length to be 34–82% [3–6], while a meta-analysis from six independent family-based cohorts estimated the heritability to be 70% (95% CI 64%–76%) [7]. These estimates from family-based studies reflect both the genetic contribution and shared environmental factors due to the relatedness in the study participants, as well as a shared intrauterine environment in some cases, and therefore are likely to be higher than the true portion of telomere length variation determined by genetic inheritance [8]. Telomere length genetic inheritance includes non-telomere region determined by genetic variance as well as direct transmission of the telomere ends from the parental gametes to the zygote, and is therefore only partially determined by gene variants [9, 10]. A recent study using genome-wide complex trait analysis (GCTA) estimated that additive genetic variance, that is, the totality of single nucleotide polymorphisms, contributed only

28% of total phenotypic variance of TL in a European American sample [11]. However, it should be noted that this study used salivary DNA for TL measurements, which are less robust than DNA from leukocytes. Yet, as of now, only several dozen single nucleotide polymorphisms (SNPs) associated with phenotypically measured telomere length have been reported [12–37]. Each of these SNPs contributes to a very small percentage of the telomere length variance [15, 38]. Therefore, our current understanding of genetic determinants of telomere length is still very limited.

Nevertheless, Mendelian randomization studies restricted to utilization of genetic scores composed of several TL-associated SNPs as instrument variables had provided strong evidence for causal links between TL and several major diseases. The Mendelian randomization study design is less susceptible to confounding and reverse causation than phenotypically measured telomere length (mTL), which is impacted by lifestyle and other environmental factors. The relationship between mTL and cancer clearly illustrates this point. Observational studies have reported associations of both short and long phenotypically measured telomere length (mTL) with cancers in various study design settings [39–42]. This complex relationship between mTL and cancer may be due to the different roles telomere length play in cancer at various stages of the pathogenesis [43] and the confounding effects of disease progression and treatment on telomere length.

Mendelian randomization designs have commonly implicated longer genetically predicted telomere length in increased risk for several cancers [44–49], while shorter genetically predicted telomere length is associated with increased rates of degenerative disease, such as cardiovascular disease, diabetes, COPD, and Alzheimer's disease [49–52]. It has been proposed that the opposing directions of these associations imply that human telomere length has evolved to balance the disease risks imposed by both short and long telomeres [53, 54], and thereby achieving an optimal length over successive generations. It would seem that this argument fails to consider the fact that the manifestation of the majority of the diseases associated with long or short telomeres, with the exception of a number of rare cancers (e. g. neuroblastoma and testicular cancer, some types of leukemia and lymphoma and type I diabetes mellitus), happens in later life, well past the reproductive period to have real evolutional pressure on the population. However, this finding among contemporary humans could alternatively suggest that selection forces have acted on telomere length precisely to lessen the impact of degenerative and neoplastic diseases during the reproductive years [55, 56].

These varying directions of disease risk beg the question: what is the relative impact of long gTL vs. short gTL on disease incidence and burden in a population? We sought to determine the contribution of genetically predicted telomere length to disease burden, including DALYs, in a European population for which relevant data were available. We defined gTL as classically genetically determined telomeres (assessed using SNPs). Here, a genetic sum score which included up to 16 telomere length associated SNPs was used as the proxy for gTL. This analysis did not take into account possible contributions from telomeres directly transmitted via parental germlines.

In the most extensive analysis published thus far, the recent meta-analysis by Haycock et al. examined the relationship between genetically determined telomere length (gTL) and 35 cancer types and 48 non-neoplastic diseases using a Mendelian Randomization design. Up to 16 TL-associated SNPs were used to approximate gTL in 420,081 cases and 1,093,105 controls from 45 published cohorts and one unpublished study. They found that 9 cancers (endometrial, ovarian serous LMP, testicular, bladder, kidney, lung adenocarcinoma, malignant skin melanoma, glioma, neuroblastoma) are associated with longer gTL, while 6 non-neoplastic diseases (coronary heart disease, aortic aneurysm, Alzheimer disease and other dementias, type I diabetes mellitus, interstitial lung disease and Celiac disease) are associated with shorter

gTL. These results confirm and extend the notion that there exists an evolutionary tradeoff in the regulation of telomere length between cancer and degenerative disease [54].

However, their results, which are presented as odds ratios (ORs), also suggest associations of markedly larger magnitude between gTL and cancers than between gTL and degenerative diseases. For instance, glioma has an OR of 5.27 per SD increase in gTL, while coronary heart disease has on OR of only 0.78 per SD increase in gTL (corresponding to an OR of 1.28 per SD shorter gTL). This imbalanced pattern of effect would seem to suggest that long telomeres have a far larger impact on incidences of cancer, and therefore has far greater clinical significance to cancer, than shorter telomeres do to non-neoplastic diseases. It perhaps even challenges the utility of telomere length as a biomarker of aging, as it has been described in a body of work [2, 57].

However, heart disease and Alzheimer's disease are common conditions with high incidence rates, while the cancers associated with long telomeres are much less common. These conditions also have vastly different diseases courses and prognoses. As a result, it is unclear what the overall impact of telomere length is on the burden of disease experienced both by individuals and by populations. To this end, we propose a quantitative approach which takes into account the increased ORs of disease associated with TL, the incidence of specific diseases, and the population burden associated with those diseases in order to explore the role of telomeres in the health of modern populations. We used incidence and Disability-Adjusted Life Years (DALY) data published by the Institute of Health Metrics and Evaluation (IHME) at the University of Washington to consider how the disease risks associated with gTL telomere length manifest in a European population. DALYs can be thought of as years of 'healthy life' lost due to disease. They are calculated as the sum of two measures: Years of Life Lost (YLL), which reflects premature mortality attributable to disease, and Years Lost due to Disability (YLD), which reflects the burden of the disease experienced by the patient prior to death. Unlike prevalence, incidence, and mortality, DALYs allow for the direct quantitative comparison of diseases with vastly different trajectories. For instance, the burden of non-fatal chronic diseases such as cardiovascular disease can be directly compared to that of rapidly fatal diseases such as aggressive cancers. Our analysis provided quantitative estimates of the impact of short and long gTL in a human population for the first time.

## Materials and methods

### Calculating excess disease incidence and burden associated with gTL

Odds ratios (ORs) of disease associated with gTL telomere length were taken from the 2017 Haycock et al. meta-analysis, which identified 16 SNPs as a genetic proxy for telomere length in order to assess the relationship with 56 primary-outcome diseases in a Mendelian Randomization design [49]. An additional 7 ORs of disease were taken from an original publication by Li et al. [38], in which ORs of disease were calculated for 122 diseases using 52 independent variants. Only odds ratios with statistical significance were included in the present analysis.

Age-standardized Disability-Adjusted Life Years (DALY) and incidence data for the 2017 year were downloaded through the GBD Results Tool (http://ghdx.healthdata.org/gbd-results-tool). Only the "Europe" population was included in order to maximize consistency with the Haycock et al. meta-analysis, where all participants were of European ancestry.

In several cancers, Haycock et al. report odds ratios for narrower disease definitions than the available IHME data. In order to most accurately approximate the excess impact of these conditions associated with gTL, we used data from the US National Cancer Institutes' publicly available SEER Database (seer.cancer.gov) to interpolate the incidence and DALYs for the conditions defined by Haycock et al. Case counts were pulled from the SEER database for both the specific cancer (e.g., squamous cell carcinoma) and for the broader category defined by the

IHME (e.g., trachea, bronchus, and lung cancers), and a percentage of total cases calculated. The IHME data were multiplied by the percentage to yield incidence and DALY values for the narrow disease definitions (See S1 File).

The ORs in Haycock et al. are all given relative to longer gTL. To allow for our calculations, the ORs for non-neoplastic diseases were inversed in order to provide an OR relative to shorter gTL. The ORs in Li et al. are given relative to shorter gTL, and are all less than 1; they were also inversed to allow for our calculations.

To capture the excess DALYs per 100,000 persons associated with gTL, the formula "(OR-1) x DALY" was applied. In other words, if an OR = 1 reflects baseline odds of disease, we subtract OR-1 to capture odds of disease predicted by gTL in excess of baseline. This value is multiplied by the corresponding DALY to estimate the excess disease burden experienced by the European population due to longer gTL. Excess incidence and 95% confidence intervals (CI) for our estimates were calculated by the same method.

## Excluded odds ratios

Celiac disease and abdominal aortic aneurysm were reported in Haycock et al. as associated with shorter gTL, but were excluded from our analysis because corresponding IHME was not available. Similarly, uterine polyps and hypothyroidism from Li et al. were excluded.

## SNP literature search

A list of all SNPs reported to be associated with TL in the published literature was compiled as follows: 1) A PubMed search for the term ["telomere length" AND (SNP OR "polymorphism, single nucleotide") AND human] was completed on November 26, 2019, yielding 211 results. All abstracts were reviewed, and studies were selected if they included a direct measure of association between peripheral blood leukocyte TL and SNPs. Selected studies include both GWAS and hypothesis-driven tests of association. Participants must have been cancer-free at the time of LTL measurement. Publications were excluded if: N <100; study design and/or statistical methods were considered as unreliable; effect allele could not be determined from data presented in publication; errors/inconsistencies were identified in the publication that precluded interpretation. 2) An additional 2 studies were identified in the GWAS Catalog in November 2019. 3) All 16 SNPs reported by Haycock et al. [49] were included, as well as any other significant SNPs in the original publications used for meta-analysis. These were captured by the above literature search methods. An additional publication by Mangino et al. [35] was identified by focused review of publications by the same authors.

Chromosomal position data for all SNPs was drawn from the PubMed SNP database, using assembly GRCh38.p7. "Short Allele" refers to the short telomere allele and is reported as in original publications. 'Long Allele' is reported as in the publication where available, or otherwise it reflects the alternative allele reported in the PubMed SNP database. All alleles are reported in forward orientation, modifying from the original publications where necessary. Short Allele frequencies are based on the '1000Genomes' global population data, found in the PubMed SNP database.

## LD determination

Linkage disequilibrium (LD) between SNPs on the same chromosome is given as $R^2$ values, obtained from the 'LDlink' suite of applications, provided by National Cancer Institute Division of Cancer Epidemiology & Genetics (https://ldlink.nci.nih.gov). The 'LDmatrix' tool was used to generate tables of $R^2$ values for all chromosomes with at least two identified SNPs. While there are a number of exceptions, the majority of GWAS studies listed in Table 1 were

**Table 1. Population health burden associated with telomere length based on Haycock et al 2017 meta-analysis.**

| | OR of Disease per SD Longer gTL (95%CI) | Incidence Rate | Excess Incidence Per 1 SD Longer gTL | Total Excess Incidence (95% CI) | DALY | Excess DALY Per 1 SD Longer gTL | Total Excess DALY (95% CI) |
|---|---|---|---|---|---|---|---|
| | | Per 100,000 Persons Age Adjusted | Per 100,000 Persons (95%CI) | | Per 100,000 Persons Age Adjusted | Per 100,000 Persons (95% CI) | |
| **Diseases Associated with Long gTL** | | | | 94.04 (45.49–168.84) | | | 1,255.25 (662.71–2,163.63) |
| Lung Adenocarcinoma | **3.19 (2.40–4.22)** | 11.56 (11.22–11.86) | 25.31 (16.18–37.22) | | 222.74 (217.17–227.96) | 487.79 (311.83–717.21) | |
| Malignant skin melanoma | **1.87 (1.55–2.26)** | 11.60 (8.27–13.33) | 10.09 (6.38–14.62) | | 53.25 (39.55–63.18) | 46.33 (29.29–67.10) | |
| Endometrial cancer | **1.31 (1.07–1.61)** | 6.23 (6.49–5.97) | 1.93 (0.44–3.80) | | 23.82 (22.62–25.02) | 7.38 (1.67–14.53) | |
| Ovary cancer (serous LMP) | **4.35 (2.39–7.94)** | 0.04 (0.03–0.04) | 0.12 (0.05–0.25) | | 0.55 (0.53–0.57) | 1.85 (0.77–3.84) | |
| Testicular germ-cell cancer | **1.76 (1.02–3.04)** | 3.05 (2.84–3.27) | 2.32 (0.06–6.22) | | 8.84 (8.20–9.58) | 6.72 (0.18–18.03) | |
| Bladder cancer | **2.19 (1.32–3.66)** | 11.26 (10.84–11.64) | 13.40 (3.60–29.95) | | 72.93 (70.02–75.87) | 86.79 (23.34–193.99) | |
| Glioma | **5.27 (3.15–8.81)** | 8.39 (7.29–9.30) | 35.83 (18.04–65.54) | | 134.46 119.42–150.86) | 574.14 (289.09–1,050.12) | |
| Neuroblastoma | **2.98 (1.92–4.62)** | 0.012 (0.010–0.013) | 0.02 (0.01–0.04) | | 0.18 (0.16–0.21) | 0.37 (0.17–0.67) | |
| Kidney cancer | **1.55 (1.08–2.23)** | 9.11 (8.39–9.48) | 5.01 (0.73–11.21) | | 79.79 (74.51–82.74) | 43.88 (6.38–98.14) | |
| | OR of Disease per SD Shorter gTL (95%CI) | Incidence Rate | Excess Incidence Per 1 SD Shorter gTL | Total Excess Incidence (95% CI) | DALY | Excess DALY Per 1 SD Shorter gTL | Total Excess DALY (95% CI) |
| | | Per 100,000 Persons Age Adjusted | Per 100,000 Persons (95%CI) | | Per 100,000 Persons Age Adjusted | Per 100,000 Persons (95% CI) | |
| **Diseases Associated with Short gTL** | | | | 121.49 (48.40–228.58) | | | 1,007.75 (411.63–1,847.34) |
| Coronary heart disease | **1.28 (1.11–1.49)** | 190.81 (171.96–211.00) | 53.43 (20.99–93.50) | | 2106.22 (2060.06–2167.79) | 589.74 (231.68–1,032.05) | |
| Aortic aneurysm | **1.59 (1.23–2.04)** | NA | NA | | 57.09 (55.47–58.72) | 33.68 (13.13–59.37) | |
| Alzheimer disease and other dementias | **1.19 (1.02–1.41)** | 101.07 (90.31–112.58) | 19.20 (2.02–41.44) | | 410.16 (382.77–438.18) | 77.93 (8.20–168.17) | |
| Type I Diabetes Mellitus | **1.41 (1.02–1.96)** | 9.59 (8.70–10.59) | 3.93 (0.19–9.21) | | 62.99 (55.19–71.48) | 25.83 (1.26–60.47) | |
| Interstitial lung disease | **11.11 (6.67–20.00)** | 4.44 (4.08–4.82) | 44.93 (25.20–84.44) | | 27.75 (22.96–31.78) | 280.57 (157.35–527.28) | |

Odds ratios represent published calculations from a GWAS meta-analysis of associations between genetically-predicted telomere length and disease by Haycock et al., 2017. In some cases, several ORs are given for narrow disease definitions within a single DALY category. These ORs were weighted by their relative incidence or prevalence (IBD, diabetes mellitus) and collapsed into a single value per DALY. Population health burden associated with telomere length was defined as "excess" odds of disease (OR-1) associated with longer telomeres multiplied by the respective DALY. Positive values reflect excess burden associated with longer TL, and negative values reflect burden in the population that has been prevented by longer TL. Disability-adjusted life years (DALYs) represent data collected for persons of all ages in the United States during the years 2000–2016, as reported by the World Health Organization (2018). ICD10 codes corresponding to each DALY disease definition are presented.

based on European populations. In order to maximize consistency with GWAS data, LD was based on five European populations: Utah Residents from North and West Europe, Toscani in Italia, Finnish in Finland, British in England and Scotland, and Iberian population in Spain.

When estimating the number of independent potential sentinel/causal variants, we consider any pair of SNPs with $R^2 < 0.5$ as uncorrelated, therefore count them as independent potential sentinel/causal variants.

## Results and discussion

On the face of it, odds ratios alone suggest that long gTL is associated with a much greater excess risk of neoplastic disease than short gTL is with excess risk of degenerative disease. However, taking into account the actual disease incidence tells a different story: the incidences of some of these degenerative conditions–namely coronary heart disease and Alzheimer's disease–far outweigh those of the neoplastic diseases (Table 1) and have disease trajectories that may be on the order of decades rather than months-to-years. When we apply our method to estimate the excess incidence of these conditions, we find that, despite markedly larger ORs of neoplastic disease, the large incidence of degenerative diseases causes the excess incidence attributable to gTL to balance or exceed that of the neoplastic diseases. For instance, the largest OR of any disease associated with gTL is that of glioma, at 5.27 (3.15–8.81, 95%CI) per SD long gTL. However, with an incidence of only 8.39 per 100,000 persons (7.29–9.30, 95%CI), the excess incidence per SD long gTL per 100,000 persons amounts to only 35.83 cases. In contrast, coronary heart disease, which is the most common condition examined at an incidence of 190.81 per 100,000 persons (171.96–211.00, 95%CI) but with an OR of only 1.28 per SD short gTL, contributes an excess incidence of 53.43 cases (20.99–93.50, 95%CI) per SD short gTL. When we combine all 9 neoplastic diseases together, long gTL is associated with a total excess incidence of 94.04 cases (45.49–168.84, 95%CI; per SD long gTL per 100,000 persons), while short gTL is associated with a total excess incidence of 121.49 cases (48.40–228.58, 95% CI; per SD short gTL per100,000 persons) from the 4 non-neoplastic diseases.

When the same method is applied to DALYs in order to assess the impact of gTL on total population health burden due to both death and disability, as opposed to incidence alone, a similar pattern is observed. Coronary heart disease is again associated with the greatest impact of gTL, with 589.74 (231.68–1032.05, 95%CI) excess DALY per SD short gTL per 100,000 persons (Table 1). However, glioma and lung adenocarcinoma follow closely behind, with 574.14 (289.09–1050.12, 95%CI; per SD long gTL per 100,000 persons) and 487.79 (311.83–717.21, 95%CI; per SD long gTL per 100,000 persons) excess DALYs, respectively. These values are the result of their large ORs of disease, as noted, but also of their large DALY burdens relative to their incidence rates. Overall, long gTL is associated with excess DALYs totaling 1,255.25 (662.71–2,163.63, 95%CI; per SD long gTL per 100,000 persons) from the 9 neoplastic diseases, while short gTL is associated with excess DALYs totaling 1,007.75 (411.63–1847.34, 95%CI; per 1SD short gTL per 100,000 persons) from the 4 non-neoplastic diseases.

Notably, interstitial lung disease contributes significantly to total excess incidence and excess DALYs associated with short gTL in the Haycock et al. meta-analysis. Idiopathic pulmonary fibrosis (IPF) likely contributes to a large portion of the burden of interstitial lung disease associated with telomeres. It is a rare condition of unknown etiology, with a prevalence of only 2-29/100,000 persons [58]. Evidence from rodent models and rare human genetics diseases (telomere syndromes) suggest that mutations in telomere maintenance genes may be directly involved in the pathogenesis of IPF [59–61], accounting for the large OR associated with the disease. Data from the Danish National Registry of Patients shows that IPF accounts for approximately 26.8% of interstitial lung disease cases, based on the IHME disease definition

[62]. If the approximated contribution of IPF is subtracted from our calculations, the total excess incidence of non-neoplastic disease associated with short gTL falls to 109.46, and the excess DALYs fall to 932.64. In other words, even when we subtract out the contribution of a disease where the pathogenesis is known to be directly related to mutations in telomere maintenance genes, the same patterns of relative contribution to disease incidence and DALYs persist.

Repeating our analysis on another recently published Mendelian Randomization study using 52 SNPs associated with LTL and health data from the UK Biobank reveals a similar pattern of contribution to the incidence and burden of neoplastic disease [38]. The Li et al. study identified five cancer types (thyroid cancer, lymphoma and multiple myeloma, leukemia, lung cancer, skin cancer (including melanoma)) and two benign conditions of abnormal cellular proliferation (uterine fibroid, benign prostatic hyperplasia (BPH)) that were significantly associated with long gTL. Based on the reported ORs, long gTL contributed a total 231.42 excess cases per 1 SD long gTL per 100,000 persons, and a total 840.28 excess DALYs per 1 SD long gTL per 100,000 persons (Table 2). The main drivers of excess incidence were uterine fibroids and BPH, with 77.71 and 62.35 excess cases per 1 SD long gTL per 100,000 persons, respectively. The contribution of gTL to cancer incidence is small in comparison, ranging from 8.18 excess cases for thyroid cancer to 37.64 excess cases for skin cancer. In contrast, the excess DALYs were overwhelmingly driven by lung cancer, with 511.26 excess DALYs per 1 SD long gTL per 100,000 persons. Leukemia contributes an additional 151.32 excess DALYs, while uterine fibroids and BPH trail far behind with only 10.71 and 19.61 DALYs, respectively. These results again demonstrate that while long gTL contributes relatively little to the population incidence of cancers, the severe morbidity and mortality associated with some cancers results in a heavy DALY burden. Also of note, Li et al. use a different and much larger set of 52 SNPs are a proxy for gTL, and with the exception of lung cancers and skin cancers, identify non-overlapping associations of gTL with disease. In the cases of lung cancers and skin cancers, the two groups use different disease definitions, with Li et al. adopting broader disease

**Table 2. Population health burden associated with telomere length based on Li et al 2020 meta-analysis.**

| Diseases Associated with Long gTL in Li et al., 2020 | OR of Disease per SD Longer gTL (95%CI) | Incidence Rate | Excess Incidence Per 1 SD Longer gTL | Total Excess Incidence (95% CI) | DALY | Excess DALY Per 1 SD Longer gTL | Total Excess DALY (95% CI) |
|---|---|---|---|---|---|---|---|
| | | Per 100,000 Persons Age Adjusted | Per 100,000 Persons (95%CI) | | Per 100,000 Persons Age Adjusted | Per 100,000 Persons (95% CI) | |
| Thyroid cancer | 2.83 | 4.46 (4.27–4.7) | 8.18 (7.83–8.62) | 231.42 (192.2–277.86) | 11.97 (11.12–13.04) | 21.93 (20.39–23.9) | 840.28 (795.6–890.32) |
| Lymphomas and multiple myeloma | 1.66 | 16.32 (15.31–18.24) | 10.69 (10.03–11.95) | | 144.59 (135.83–158.96) | 94.72 (88.99–104.14) | |
| Uterine fibroid | 1.66 | 118.03 (89.95–152.97) | 77.71 (59.23–100.72) | | 16.27 (9–27.78) | 10.71 (5.92–18.29) | |
| Benign prostatic hyperplasia (BPH) | 1.46 | 135.73 (121.06–150.51) | 62.35 (55.62–69.14) | | 42.68 (27.74–60.73) | 19.61 (12.74–27.9) | |
| Leukemia | 2.01 | 8.24 (7.82–8.62) | 8.3 (7.88–8.68) | | 150.16 (143.66–156.15) | 151.32 (144.77–157.36) | |
| Lung cancer | 1.81 | 32.57 (31.62–33.41) | 26.53 (25.77–27.22) | | 627.59 (611.9–642.32) | 511.36 (498.57–523.36) | |
| Skin cancer (including melanoma) | 1.45 | 84.05 (57.7–115.05) | 37.64 (25.84–51.53) | | 68.37 (54.09–78.98) | 30.62 (24.22–35.37) | |

A process similar to that of Table 1 was applied to Li et al and the results presented.

categories in both cases. This highlights the need for standardized methods of SNP selection and case definition in the Mendelian Randomization studies of telomere length to facilitate meaningful replication and extrapolation.

These results suggest that genetically-determined long and short telomere lengths are associated with disease burden of approximately equal magnitude, despite their vastly different ORs. Short gTL is associated with slightly higher disease incidences, and long gTL with slightly higher DALYs. We believe these results provide the first quantitative estimate of the relative impacts of short and long telomere length on the health of a human population. Our results are also consistent with earlier reports that human telomere length is regulated under opposing evolutionary forces that act to minimize the risks of both neoplastic and non-neoplastic diseases [53].

However, our results should be interpreted with caution. They are based on a calculated measure of genetically-predicted TL using only 16 SNPs as a proxy by Haycock et al. These SNPs only account for a small percentage of the large telomere length variation in the population. The estimation of 28% narrow inheritance with genome-wide SNP data suggest that it is likely that more SNPs with small effect size are yet to be discovered [11]. How the totality of the all TL SNPs contribute to disease burden is unknown. We compiled an updated, comprehensive list of all SNPs associated with measured leukocyte telomere length in the published literature (Table 3), and assessed linkage disequilibrium (LD) between nearby sites (Table 4). LD between two SNPs in the same gene suggests a common causal genetic variant shared between the two loci, while SNPs not in LD suggest two distinct genetic causes of variance in TL. This list includes 106 SNPs on 18 chromosomes. Linkage disequilibrium analysis (Table 4) suggests that these 106 SNPs likely reflect 70 distinct causal variants from 50 genes, only 18 of which are known to be mechanistically involved in telomere maintenance pathways (Table 3). This list provides a tool for future studies of Mendelian Randomization using these SNPs as proxies for genetically determined telomere length. However, caution should still be applied when using these SNPs for future studies. Mendelian randomization approaches are based on the assumptions that the (1) selected SNPs are associated with telomere length; (2) the selected SNPs are not associated with confounders; and (3) the selected SNPs are associated with disease exclusively through their effect on telomere length. Therefore, the candidate SNPs to be used in MR studies should be from those genes that have been well-documented as mechanistically involved in telomere maintenance (Table 4). Even with this caution, we note that some telomere maintenance genes have functions other than telomere length. For example, in addition to extending telomeres, telomerase protein gene *hTERT* is also reported to be involved in NF-kb and Wnt/b-catenin transcriptional pathways [63, 64] and is localized to mitochondria to inhibit caspase mediated apoptosis [65, 66]. Similarly, it is possible that genes with yet unknown functions will have both telomere and non-telomere functions.

We also note several other limitations. First, genetic determinants of telomere length include both the variation in the non-telomeric regions attributable to SNPs and the direct inheritance of the lengths of telomeres from the oocyte and sperm when the zygote was formed. A recent paper examining the extent of physical telomere sharing among relatives suggests that the direct transmission of telomeres from gametes to zygotes contribute to at least 11% of the telomere length variability [9]. The mechanisms of how the telomere lengths of the oocyte and sperm contribute to the zygote, and how the initial length of telomeres is reset after fertilization is largely unknown [67]. Telomere length of newborns, which likely reflects the impact of the genetic determinants and the prenatal environment, will play an important role in contributing to the risks of both neoplastic and non-neoplastic disease in adult life [68].

Second, as environmental factors can influence telomere length throughout the whole lifespan, and perhaps differentially during different developmental stages (childhood, adulthood

**Table 3. List of SNPs associated with telomere length.**

| Telomere Maintenance Gene | SNP | Chr. | Position (GRCh38. p7) | Number of Participants | GWAS | Population | Paper | short allele | long allele | Short Allele Frequency (1000 Genomes) |
|---|---|---|---|---|---|---|---|---|---|---|
| N | rs621559 | 1 | 43179740 | 1,619 | Y | Texas, no ethnicity criteria | Gu et al., 2011 | G | A | 0.908 |
| | | | | 550 | N | Chinese | Shi et al., 2013 | | | |
| Y | rs3219104 | 1 | 226374920 | 23,096 | Y | Singapore Chinese | Dorajoo et al., 2019 | A | C | 0.183 |
| | | | | 78,592 | Y | European | Li et al., 2020 | | | |
| N | rs1805087 | 1 | 236885200 | 989 | N | Non-Hispanic White Female | Kim et al., 2012 | A | G | 0.804 |
| N | rs11125529 | 2 | 54248729 | 37,684 | Y | European | Codd et al., 2013 | C | A | 0.875 |
| | rs11890390 | 2 | 54258545 | 60,061 | Y | Singapore Chinese European | Dorajoo et al., 2019 | C | T | 0.858 |
| N | rs6772228 | 3 | 58390292 | 26,089 | Y | European | Pooley et al., 2013 | A | T | 0.056 |
| Y | rs55749605 | 3 | 101513249 | 78,592 | Y | European | Li et al., 2020 | A | C | 0.637 |
| Y | rs12638862 | 3 | 169759718 | 5,075 | Y | Bangladeshi | Delgado et al., 2018 | G | A | 0.272 |
| | | | | 4,289 | N | Caucasian (American + Danish) w/ Familial Longevity | Lee et al., 2014 | | | |
| | rs12696304 | 3 | 169763483 | 9,492 | Y | European | Codd et al., 2010 | G | C | 0.281 |
| | | | | 470 | N | Arab (Kuwaiti) Healthy + T2DM | Al Khaldi et al., 2015 | | | |
| | | | | 4,016 | N | Han Chinese | Shen et al., 2011 | | | |
| | | | | 1,002 | N | European (Spain) w/ CHD | Gomez-Delgado et al., 2018 | | | |
| | rs2293607 | 3 | 169764547 | 2,953 | N | American, European | Njajou et al., 2010 | C | T | 0.249 |
| | | | | 23,096 | Y | Singapore Chinese | Dorajoo et al., 2019 | | | |
| | rs10936599 | 3 | 169774313 | 37,684 | Y | European | Codd et al., 2013 | T | C | 0.245 |
| | | | | 4,289 | N | Caucasian (American + Danish) w/ Familial Longevity | Lee et al., 2014 | | | |
| | rs1317082 | 3 | 169779797 | 9,190 | Y | European | Mangino et al., 2012 | G | A | 0.249 |
| | | | | 26,089 | Y | European | Pooley et al., 2013 | | | |
| | | | | 4,289 | N | Caucasian (American + Danish) w/ Familial Longevity | Lee et al., 2014 | | | |
| | rs3772190 | 3 | 169782699 | 3,417 | N | European African-American | Levy et al., 2010 | A | G | 0.147 |
| | rs10936600 | 3 | 169796797 | 78,592 | Y | European | Li et al., 2020 | T | A | 0.229 |
| | rs16847897 | 3 | 169850328 | 9,492 | Y | European | Codd et al., 2010 | C | G | 0.282 |
| | | | | 470 | N | Arab (Kuwaiti) Healthy + T2DM | Al Khaldi et al., 2015 | | | |
| | | | | 4,016 | N | Han Chinese | Shen et al., 2011 | | | |
| | | | | 3,554 | N | European | Prescott et al., 2011 | | | |
| | rs1920116 | 3 | 169862183 | 37,684 | Y | European | Walsh et al., 2014 | A | G | 0.285 |
| N | rs13137667 | 4 | 70908630 | 78,592 | Y | European | Li et al., 2020 | T | C | 0.032 |
| N | rs7680468 | 4 | 107383042 | 4,289 | Y | Caucasian (American + Danish) w/ Familial Longevity | Lee et al., 2014 | T | G | 0.031 |

*(Continued)*

**Table 3.** (Continued)

| Telomere Maintenance Gene | SNP | Chr. | Position (GRCh38. p7) | Number of Participants | GWAS | Population | Paper | short allele | long allele | Short Allele Frequency (1000 Genomes) |
|---|---|---|---|---|---|---|---|---|---|---|
| Y | rs7675998 | 4 | 163086668 | 37,684 | Y | European | Codd et al., 2013 | A | G | 0.227 |
| | rs4691895 | 4 | 163127047 | 78,592 | Y | European | Li et al., 2020 | G | C | 0.236 |
| | rs10857352 | 4 | 163180330 | 23,096 | Y | Singapore Chinese | Dorajoo et al., 2019 | A | G | 0.581 |
| Y | rs2075786 | 5 | 1266195 | 207 | N | Chinese w/ Schizophrenia + Healthy | Rao et al., 2016 | G | A | 0.640 |
| | rs10054203 | 5 | 1279849 | 774 | N | Chinese | Li et al., 2019 | G | C | 0.571 |
| | rs7726159 | 5 | 1282204 | 26,089 | Y | European | Pooley et al., 2013 | C | A | 0.694 |
| | | | | 774 | N | Chinese | Li et al., 2019 | | | |
| | rs7705526 | 5 | 1285859 | 5,075 | Y | Bangladeshi | Delgado et al., 2018 | C | A | 0.660 |
| | | | | 15,567 | N | European | Bojesen et al., 2013 | | | |
| | | | | 23,096 | Y | Singapore Chinese | Dorajoo et al., 2019 | | | |
| | rs2736100 | 5 | 1286401 | 6,549 | Y | Han Chinese (Healthy + T2DM) | Liu et al., 2014 | A | C | 0.496 |
| | | | | 37,684 | Y | European | Walsh et al., 2014 | | | |
| | | | | 37,684 | Y | European | Codd et al., 2013 | | | |
| | | | | 913 | N | Han Chinese | Gu et al., 2016 | | | |
| | | | | 390 | N | Chinese Women | Lan et al., 2013 | | | |
| | rs2853677 | 5 | 1287079 | 78,592 | Y | European | Li et al., 2020 | A | G | 0.586 |
| | rs2736108 | 5 | 1297373 | 15,567 | N | European | Bojesen et al., 2013 | C | T | 0.733 |
| | rs401681 | 5 | 1321972 | 1,208 | N | Caucasian | Bao et al., 2017 | T | C | 0.433 |
| N | rs2966952 | 5 | 7867917 | 989 | N | Non-Hispanic White Female | Kim et al., 2012 | T | C | 0.169 |
| N | rs3733890 | 5 | 79126136 | 989 | N | Non-Hispanic White Female | Kim et al., 2012 | A | G | 0.304 |
| N | rs34991172 | 6 | 25480100 | 78,592 | Y | European | Li et al., 2020 | G | T | 0.073 |
| N | rs1800629 | 6 | 31575254 | 840 | N | European (Spain) w/ CHD | Rangel-Zuniga et al., 2016 | G | A | 0.845 |
| N | rs2736176 | 6 | 31619784 | 78,592 | Y | European | Li et al., 2020 | G | C | 0.696 |
| N | rs558702 | 6 | 31902549 | 989 | N | Non-Hispanic White Female | Kim et al., 2012 | A | G | 0.097 |
| N | rs654128 | 6 | 116765215 | 1619 | Y | Texas, no ethnicity criteria | Gu et al., 2011 | C | A | 0.846 |
| Y | rs59294613 | 7 | 124914213 | 78,592 | Y | European | Li et al., 2020 | A | C | 0.291 |
| | rs7776744 | 7 | 124959695 | 23,096 | Y | Singapore Chinese | Dorajoo et al., 2019 | G | A | 0.590 |
| Y | rs11991621 | 8 | 9549072 | 3,646 | N | Non-Hispanic White (M) Caucasian (F) | Mirabello et al., 2010 | T | C | 0.164 |
| | rs6990097 | 8 | 9555347 | | N | Swedish Female | Varadi et al., 2009 | T | C | 0.740 |
| | rs12549064 | 8 | 9584517 | 3,646 | N | Non-Hispanic White (M) Caucasian (F) | Mirabello et al., 2010 | C | A | 0.171 |
| | rs10903314 | 8 | 9609596 | 3,646 | N | Non-Hispanic White (M) Caucasian (F) | Mirabello et al., 2010 | T | C | 0.258 |
| | rs6990300 | 8 | 9690351 | 3,646 | N | Non-Hispanic White (M) Caucasian (F) | Mirabello et al., 2010 | G | A | 0.334 |
| | rs11249943 | 8 | 9750353 | 3,646 | N | Non-Hispanic White (M) Caucasian (F) | Mirabello et al., 2010 | C | A | 0.185 |
| | rs17150478 | 8 | 9783524 | 3,646 | N | Non-Hispanic White (M) Caucasian (F) | Mirabello et al., 2010 | G | A | 0.167 |

(*Continued*)

**Table 3.** (Continued)

| Telomere Maintenance Gene | SNP | Chr. | Position (GRCh38.p7) | Number of Participants | GWAS | Population | Paper | short allele | long allele | Short Allele Frequency (1000 Genomes) |
|---|---|---|---|---|---|---|---|---|---|---|
| Y | rs28365964 | 8 | 73008648 | 23,096 | Y | Singapore Chinese | Dorajoo et al., 2019 | T | C | 0.999 |
| NA | rs10466239 | 10 | 43354379 | 4,289 | N | Caucasian (American + Danish) w/ Familial Longevity | Lee et al., 2014 | C | T | 0.918 |
| N | rs7095953 | 10 | 99514668 | 60,061 | Y | Singapore Chinese European | Dorajoo et al., 2019 | C | T | 0.267 |
| Y | rs7100920 | 10 | 103881220 | 4,289 | N | Caucasian (American + Danish) w/ Familial Longevity | Lee et al., 2014 | T | C | 0.491 |
| | rs2067832 | 10 | 103883376 | 4,289 | N | Caucasian (American + Danish) w/ Familial Longevity | Lee et al., 2014 | A | G | 0.487 |
| | rs10786775 | 10 | 103897558 | 2,353 | N | European (Healthy, CHD, T2DM) | Maubaret et al., 2013 | C | G | 0.897 |
| | rs2487999 | 10 | 103900068 | 26,089 | Y | European | Pooley et al., 2013 | C | T | 0.896 |
| | rs9419958 | 10 | 103916188 | 9,190 | Y | European | Mangino et al., 2012 | C | T | 0.852 |
| | rs9420907 | 10 | 103916707 | 37,684 | Y | European | Codd et al., 2013 | A | C | 0.828 |
| | rs4387287 | 10 | 103918139 | 3,417 | Y | European African-American | Levy et al., 2010 | C | A | 0.819 |
| | rs12415148 | 10 | 103920828 | 23,096 | Y | Singapore Chinese | Dorajoo et al., 2019 | T | C | 0.999 |
| Y | rs669976 | 11 | 64806117 | 3,646 | N | Non-Hispanic White (M) Caucasian (F) | Mirabello et al., 2010 | C | T | 0.102 |
| | rs524386 | 11 | 64817487 | 3,646 | N | Non-Hispanic White (M) Caucasian (F) | Mirabello et al., 2010 | C | T | 0.103 |
| | rs2957154 | 11 | 64817515 | 3,646 | N | Non-Hispanic White (M) Caucasian (F) | Mirabello et al., 2010 | C | T | 0.264 |
| Y | rs670358 | 11 | 64824207 | 3,646 | N | Non-Hispanic White (M) Caucasian (F) | Mirabello et al., 2010 | A | G | 0.118 |
| N | rs660339 | 11 | 73978059 | 950 | N | Australian Caucasian | Zhou et al., 2016 | G | A | 0.598 |
| | | | | 194 | N | Calabria, Italy; <85yrs ONLY* | Dato et al., 2017 | A | G | 0.402 |
| | rs659366 | 11 | 73983709 | 950 | N | Australian Caucasian | Zhou et al., 2016 | C | T | 0.627 |
| | | | | 569 | N | Caucasian w/ T2DM | Salpea et al., 2010 | T | C | 0.373 |
| | | | | 194 | N | Calabria, Italy; <85yrs ONLY* | Dato et al., 2017 | | | |
| Y | rs12270338 | 11 | 94414298 | 3,646 | N | Non-Hispanic White (M) Caucasian (F) | Mirabello et al., 2010 | C | A | 0.791 |
| | rs13447720 | 11 | 94432160 | 3,646 | N | Non-Hispanic White (M) Caucasian (F) | Mirabello et al., 2010 | T | C | 0.784 |
| Y | rs1801516 | 11 | 70025996 | 989 | N | Non-Hispanic White Female | Kim et al., 2012 | G | A | 0.865 |
| | rs228595 | 11 | 108234866 | 78,592 | Y | European | Li et al., 2020 | A | G | 0.427 |
| | rs227080 | 11 | 108377161 | 23,096 | Y | Singapore Chinese | Dorajoo et al., 2019 | G | A | 0.529 |
| N | rs12299470 | 12 | 3528284 | 989 | N | Non-Hispanic White Female | Kim et al., 2012 | G | A | 0.873 |
| N | rs2630578 | 12 | 32152853 | 843 | N | >98.5% European w/ CAD | Mangino et al., 2008 | C | G | 0.178 |
| | rs1151026 | 12 | 32176235 | 843 | N | >98.5% European w/ CAD | Mangino et al., 2008 | G | A | 0.200 |
| N | rs17653722 | 12 | 52193734 | 6,549 | Y | Han Chinese (Healthy + T2DM) | Liu et al., 2014 | G | T | 0.833 |

(*Continued*)

**Table 3.** (Continued)

| Telomere Maintenance Gene | SNP | Chr. | Position (GRCh38.p7) | Number of Participants | GWAS | Population | Paper | short allele | long allele | Short Allele Frequency (1000 Genomes) |
|---|---|---|---|---|---|---|---|---|---|---|
| Y | rs938886 | 14 | 20369542 | 100 | N | Swedish Female | Varadi et al., 2009 | G | C | 0.761 |
| | rs4246977 | 14 | 20414432 | 100 | N | Swedish Female | Varadi et al., 2009 | C | T | 0.389 |
| Y | rs41293836 | 14 | 24252121 | 23,096 | Y | Singapore Chinese | Dorajoo et al., 2019 | C | T | 0.998 |
| N | rs1483898 | 14 | 42336702 | 492 | Y | African American Children/ Adolescents | Zeiger et al., 2018 | T | A | 0.857 |
| NA | rs398652 | 14 | 56058851 | 1619 | Y | Texas, no ethnicity criteria | Gu et al., 2011 | G | A | 0.840 |
| | | | | 550 | N | Chinese | Shi et al., 2013 | | | |
| N | rs2302588 | 14 | 72938044 | 60,061 | Y | Singapore Chinese European | Dorajoo et al., 2019 | G | C | 0.897 |
| | rs2535913 | 14 | 72948525 | 20,022 | Y | European | Mangino et al., 2015 | A | G | 0.255 |
| N | rs10046 | 15 | 51210789 | 2,143 | N | Anglo-Celtic Australian Males | Yeap et al., 2016 | G | A | 0.503 |
| | rs2899470 | 15 | 51211480 | 2,143 | N | Anglo-Celtic Australian Males | Yeap et al., 2016 | T | G | 0.407 |
| | rs700518 | 15 | 51236915 | 2,143 | N | Anglo-Celtic Australian Males | Yeap et al., 2016 | T | C | 0.527 |
| N | rs17817449 | 16 | 53779455 | 783 | N | Prague Women | Dlouha et al., 2012 | G | T | 0.398 |
| | rs9939609 | 16 | 53786615 | 1,184 | N | Korean | Yu et al., 2017 | A | T | 0.412 |
| N | rs74019828 | 16 | 58175370 | 4,013 | Y | Punjabi Sikh w/ T2DM | Saxena et al., 2014 | A | G | 0.076 |
| Y | rs3785074 | 16 | 69373083 | 78,592 | Y | European | Li et al., 2020 | A | G | 0.705 |
| N | rs62053580 | 16 | 74646176 | 78,592 | Y | European | Li et al., 2020 | G | A | 0.147 |
| N | rs7194734 | 16 | 82166375 | 78,592 | Y | European | Li et al., 2020 | T | C | 0.783 |
| | rs2967374 | 16 | 82176256 | 60,061 | Y | Singapore Chinese European | Dorajoo et al., 2019 | G | A | 0.788 |
| Y | rs3027234 | 17 | 8232774 | 11,416 | Y | European | Mangino et al., 2012 | T | C | 0.163 |
| N | rs78148049 | 17 | 64655172 | 4,289 | N | Caucasian (American + Danish) w/ Familial Longevity | Lee et al., 2014 | C/T | C/T | C = 0.961 T = 0.039 |
| Y | rs820152 | 17 | 75620008 | 3,646 | N | Non-Hispanic White (M) Caucasian (F) | Mirabello et al., 2010 | C | T | 0.372 |
| N | rs1001761 | 18 | 662103 | 60,061 | Y | Singapore Chinese European | Dorajoo et al., 2019 | A | G | 0.473 |
| NA | rs2162440 | 18 | 37634043 | 2,790 | Y | European (F) | Mangino et al., 2009 | G | A | 0.795 |
| N | rs7235755 | 18 | 37636298 | 2,790 | Y | European (F) | Mangino et al., 2009 | G | A | 0.784 |
| N | rs8105767 | 19 | 22032639 | 37,684 | Y | European | Codd et al., 2013 | A | G | 0.686 |
| N | rs7253490 | 19 | 22110904 | 60.061 | Y | Singapore Chinese European | Dorajoo et al., 2019 | C | A | 0.685 |
| N | rs412658 | 19 | 22176638 | 11,416 | Y | European | Mangino et al., 2012 | C | T | 0.646 |
| NA | rs6028466 | 20 | 39500359 | 1619 | Y | Texas, no ethnicity criteria | Gu et al., 2011 | G | A | 0.939 |
| Y | rs73598374 | 20 | 44651586 | 168 | N | Italian-Caucasian | Concetti et al., 2015 | T | C | 0.065 |

(*Continued*)

**Table 3.** (*Continued*)

| Telomere Maintenance Gene | SNP | Chr. | Position (GRCh38. p7) | Number of Participants | GWAS | Population | Paper | short allele | long allele | Short Allele Frequency (1000 Genomes) |
|---|---|---|---|---|---|---|---|---|---|---|
| Y | rs75691080 | 20 | 63638397 | 78,592 | Y | European | Li et al., 2020 | T | C | 0.100 |
| | rs34978822 | 20 | 63660246 | 78,592 | Y | European | Li et al., 2020 | G | C | 0.018 |
| | rs41309367 | 20 | 63678201 | 23,096 | Y | Singapore Chinese | Dorajoo et al., 2019 | T | C | 0.701 |
| | rs6010620 | 20 | 63678486 | 37,684 | Y | European | Walsh et al., 2014 | G | A | 0.774 |
| | rs2297439 | 20 | 63679775 | 5,075 | Y | Bangladeshi | Delgado et al., 2018 | G | T | 0.083 |
| | rs755017 | 20 | 63790269 | 37,684 | Y | European | Codd et al., 2013 | A | G | 0.874 |
| | rs73624724 | 20 | 63805045 | 78,592 | Y | European | Li et al., 2020 | T | C | 0.864 |

Information on the SNP position and the nearest gene was obtained from NCBI's SNP database https://www.ncbi.nlm.nih.gov/snp/ (GRCh3.p12). Allele frequencies were obtained from 1000 Genomes database (https://www.internationalgenome.org/data/).

**Table 4. Linkage disequilibrium between SNPs associated with telomere length.**

Chromosome 1

| RS_number | rs621559 | rs3219104 | rs1805087 | | | | | | | |
|---|---|---|---|---|---|---|---|---|---|---|
| rs621559 | 1 | | | | | | | | | |
| rs3219104 | 0 | 1 | | | | | | | | |
| rs1805087 | 0.001 | 0 | 1 | | | | | | | |

Chromosome 2

| RS_number | rs11125529 | rs11890390 | | | | | | | | |
|---|---|---|---|---|---|---|---|---|---|---|
| rs11125529 | 1 | | | | | | | | | |
| rs11890390 | 0.951 | 1 | | | | | | | | |

Chromosome 3

| RS_number | rs6772228 | rs55749605 | rs12638862 | rs12696304 | rs2293607 | rs10936599 | rs1317082 | rs3772190 | rs10936600 | rs16847897 |
|---|---|---|---|---|---|---|---|---|---|---|
| rs6772228 | 1 | | | | | | | | | |
| rs55749605 | 0 | 1 | | | | | | | | |
| rs12638862 | 0 | 0.001 | 1 | | | | | | | |
| rs12696304 | 0 | 0.002 | 0.945 | 1 | | | | | | |
| rs2293607 | 0 | 0.001 | 0.933 | 0.881 | 1 | | | | | |
| rs10936599 | 0 | 0 | 0.928 | 0.876 | 0.995 | 1 | | | | |
| rs1317082 | 0 | 0 | 0.928 | 0.876 | 0.995 | 1 | 1 | | | |
| rs3772190 | 0 | 0 | 0.928 | 0.876 | 0.995 | 1 | 1 | 1 | | |
| rs10936600 | 0 | 0 | 0.928 | 0.876 | 0.995 | 1 | 1 | 1 | 1 | |
| rs16847897 | 0 | 0.001 | 0.479 | 0.458 | 0.524 | 0.528 | 0.528 | 0.528 | 0.528 | 1 |
| rs1920116 | 0 | 0.001 | 0.479 | 0.452 | 0.524 | 0.528 | 0.528 | 0.528 | 0.528 | 0.990 |

Chromosome 4

| RS_number | rs13137667 | rs7680468 | rs7675998 | rs4691895 | rs10857352 | | | | | |
|---|---|---|---|---|---|---|---|---|---|---|
| rs13137667 | 1 | | | | | | | | | |
| rs7680468 | 0.001 | 1 | | | | | | | | |
| rs7675998 | 0.001 | 0 | 1 | | | | | | | |
| rs4691895 | 0.001 | 0 | 0.967 | 1 | | | | | | |

(*Continued*)

**Table 4.** (Continued)

| rs10857352 | 0.005 | 0 | 0.203 | 0.207 | 1 | | | | | |

**Chromosome 5**

| RS_number | rs2075786 | rs10054203 | rs7726159 | rs7705526 | rs2736100 | rs2853677 | rs2736108 | rs401681 | rs2966952 | rs3733890 |
|---|---|---|---|---|---|---|---|---|---|---|
| rs2075786 | 1 | | | | | | | | | |
| rs10054203 | 0.02 | 1 | | | | | | | | |
| rs7726159 | 0.047 | 0.607 | 1 | | | | | | | |
| rs7705526 | 0.023 | 0.454 | 0.788 | 1 | | | | | | |
| rs2736100 | 0.003 | 0.316 | 0.516 | 0.510 | 1 | | | | | |
| rs2853677 | 0 | 0.061 | 0.181 | 0.185 | 0.435 | 1 | | | | |
| rs2736108 | 0.014 | 0.001 | 0.015 | 0.042 | 0.149 | 0.220 | 1 | | | |
| rs401681 | 0.016 | 0.013 | 0.003 | 0 | 0.004 | 0.014 | 0.174 | 1 | | |
| rs2966952 | 0 | 0.001 | 0 | 0.001 | 0 | 0 | 0.001 | 0.001 | 1 | |
| rs3733890 | 0.002 | 0 | 0 | 0.001 | 0.001 | 0.001 | 0.001 | 0 | 0 | 1 |

**Chromosome 6**

| RS_number | rs34991172 | rs1800629 | rs2736176 | rs558702 | rs654128 |
|---|---|---|---|---|---|
| rs34991172 | 1 | | | | |
| rs1800629 | 0.098 | 1 | | | |
| rs2736176 | 0.012 | 0.035 | 1 | | |
| rs558702 | 0.140 | 0.450 | 0.035 | 1 | |
| rs654128 | 0.003 | 0.001 | 0.001 | 0.003 | 1 |

**Chromosome 7**

| RS_number | rs59294613 | rs7776744 |
|---|---|---|
| rs59294613 | 1 | |
| rs7776744 | 0.249 | 1 |

**Chromosome 8**

| RS_number | rs11991621 | rs6990097 | rs12549064 | rs10903314 | rs6990300 | rs11249943 | rs17150478 | rs28365964 |
|---|---|---|---|---|---|---|---|---|
| rs11991621 | 1 | | | | | | | |
| rs6990097 | 0.531 | 1 | | | | | | |
| rs12549064 | 0.808 | 0.568 | 1 | | | | | |
| rs10903314 | 0.565 | 0.747 | 0.585 | 1 | | | | |
| rs6990300 | 0.395 | 0.518 | 0.410 | 0.696 | 1 | | | |
| rs11249943 | 0.683 | 0.421 | 0.705 | 0.537 | 0.468 | 1 | | |
| rs17150478 | 0.651 | 0.378 | 0.662 | 0.488 | 0.404 | 0.763 | 1 | |
| rs28365964 | NA | NA | NA | NA | NA | NA | NA | NA |

**Chromosome 10**

| RS_number | rs10466239 | rs7095953 | rs7100920 | rs2067832 | rs10786775 | rs2487999 | rs9419958 | rs9420907 | rs4387287 | rs12415148 |
|---|---|---|---|---|---|---|---|---|---|---|
| rs10466239 | 1 | | | | | | | | | |
| rs7095953 | 0 | 1 | | | | | | | | |
| rs7100920 | 0 | 0 | 1 | | | | | | | |
| rs2067832 | 0 | 0 | 0.996 | 1 | | | | | | |
| rs10786775 | 0 | 0 | 0.113 | 0.117 | 1 | | | | | |
| rs2487999 | 0 | 0 | 0.113 | 0.117 | 1 | 1 | | | | |
| rs9419958 | 0 | 0.001 | 0.17 | 0.174 | 0.639 | 0.639 | 1 | | | |
| rs9420907 | 0 | 0.001 | 0.17 | 0.174 | 0.639 | 0.639 | 1 | 1 | | |
| rs4387287 | 0.001 | 0.001 | 0.108 | 0.111 | 0.525 | 0.525 | 0.822 | 0.822 | 1 | |
| rs12415148 | 0.015 | 0.004 | 0.009 | 0.009 | 0.001 | 0.001 | 0.002 | 0.002 | 0.002 | 1 |

**Chromosome 11**

| RS_number | rs669976 | rs524386 | rs2957154 | rs670358 | rs660339 | rs659366 | rs12270338 | rs13447720 | rs228595 | rs1801516 |
|---|---|---|---|---|---|---|---|---|---|---|

(*Continued*)

**Table 4.** (Continued)

| | | | | | | | | | | |
|---|---|---|---|---|---|---|---|---|---|---|
| rs669976 | 1 | | | | | | | | | |
| rs524386 | 0.688 | 1 | | | | | | | | |
| rs2957154 | 0.005 | 0.018 | 1 | | | | | | | |
| rs670358 | 0.155 | 0.211 | 0.001 | 1 | | | | | | |
| rs660339 | 0.001 | 0.003 | 0 | 0.001 | 1 | | | | | |
| rs659366 | 0.001 | 0.003 | 0.001 | 0.001 | 0.822 | 1 | | | | |
| rs12270338 | 0.002 | 0 | 0.002 | 0.004 | 0.002 | 0.002 | 1 | | | |
| rs13447720 | 0.001 | 0 | 0.001 | 0.006 | 0.002 | 0.002 | 0.948 | 1 | | |
| rs228595 | 0 | 0 | 0.004 | 0.007 | 0 | 0 | 0 | 0 | 1 | | |
| rs1801516 | 0 | 0.001 | 0 | 0 | 0 | 0 | 0 | 0 | 0.161 | 1 | |
| rs227080 | 0 | 0 | 0.006 | 0.003 | 0.001 | 0 | 0 | 0 | 0.460 | 0.127 | |

**Chromosome 12**

| RS_number | rs12299470 | rs2630578 | rs1151026 | rs17653722 |
|---|---|---|---|---|
| rs12299470 | 1 | | | |
| rs2630578 | 0.006 | 1 | | |
| rs1151026 | 0.004 | 0.860 | 1 | |
| rs17653722 | 0 | 0 | 0 | 1 |

**Chromosome 14**

| RS_number | rs938886 | rs4246977 | rs41293836 | rs1483898 | rs398652 | rs2302588 | rs2535913 |
|---|---|---|---|---|---|---|---|
| rs938886 | 1 | | | | | | |
| rs4246977 | 0.005 | 1 | | | | | |
| rs41293836 | 0.004 | 0.008 | 1 | | | | |
| rs1483898 | 0 | 0 | 0 | 1 | | | |
| rs398652 | 0.003 | 0.001 | 0 | 0.001 | 1 | | |
| rs2302588 | 0.001 | 0 | 0.001 | 0.001 | 0.001 | 1 | |
| rs2535913 | 0 | 0 | 0 | 0 | 0 | 0.058 | 1 |

**Chromosome 15**

| RS_number | rs10046 | rs2899470 | rs700518 |
|---|---|---|---|
| rs10046 | 1 | | |
| rs2899470 | 0.880 | 1 | |
| rs700518 | 0.837 | 0.746 | 1 |

**Chromosome 16**

| RS_number | rs17817449 | rs9939609 | rs74019828 | rs3785074 | rs62053580 | rs7194734 | rs2967374 |
|---|---|---|---|---|---|---|---|
| rs17817449 | 1 | | | | | | |
| rs9939609 | 0.996 | 1 | | | | | |
| rs74019828 | 0.002 | 0.002 | 1 | | | | |
| rs3785074 | 0.001 | 0.001 | 0.001 | 1 | | | |
| rs62053580 | 0 | 0 | 0.001 | 0 | 1 | | |
| rs7194734 | 0 | 0 | 0.001 | 0 | 0.001 | 1 | |
| rs2967374 | 0 | 0 | 0.002 | 0 | 0.001 | 0.954 | 1 |

**Chromosome 17**

| RS_number | rs3027234 | rs78148049 | rs820152 |
|---|---|---|---|
| rs3027234 | 1 | | |
| rs78148049 | 0 | 1 | |
| rs820152 | 0 | 0 | 1 |

**Chromosome 18**

| RS_number | rs1001761 | rs2162440 | rs7235755 |
|---|---|---|---|
| rs1001761 | 1 | | |

(*Continued*)

**Table 4.** (Continued)

| rs2162440 | 0 | 1 | | | | | | | | |
|---|---|---|---|---|---|---|---|---|---|---|
| rs7235755 | 0 | 1 | 1 | | | | | | | |
| **Chromosome 19** | | | | | | | | | | |
| RS_number | rs8105767 | rs7253490 | rs412658 | | | | | | | |
| rs8105767 | 1 | | | | | | | | | |
| rs7253490 | 0.821 | 1 | | | | | | | | |
| rs412658 | 0.534 | 0.603 | 1 | | | | | | | |
| **Chromosome 20** | | | | | | | | | | |
| RS_number | rs6028466 | rs73598374 | rs75691080 | rs34978822 | rs41309367 | rs6010620 | rs2297439 | rs755017 | rs73624724 | |
| rs6028466 | 1 | | | | | | | | | |
| rs73598374 | 0 | 1 | | | | | | | | |
| rs75691080 | 0.001 | 0 | 1 | | | | | | | |
| rs34978822 | 0.001 | 0 | 0.001 | 1 | | | | | | |
| rs41309367 | 0 | 0.001 | 0.042 | 0.029 | 1 | | | | | |
| rs6010620 | 0 | 0.001 | 0.026 | 0.047 | 0.621 | 1 | | | | |
| rs2297439 | 0 | 0 | 0.518 | 0.001 | 0.039 | 0.025 | 1 | | | |
| rs755017 | 0.001 | 0.001 | 0.013 | 0 | 0.204 | 0.090 | 0.011 | 1 | | |
| rs73624724 | 0.001 | 0.003 | 0.006 | 0 | 0.180 | 0.077 | 0.004 | 0.913 | 1 | |

Given as $R^2$ values, obtained from the 'LDlink' (https://ldlink.nci.nih.gov). Includes all chromosomes where at least two SNPs were in LD, based on a cutoff of $R^2 = 0.50$.

and geriatric), the disease risks caused by short or long telomere length can change accordingly. Although it should be noted that compared to the inter-individual TL variation at birth, the overall magnitude of the effect of environmental factors is smaller [69]. Adding to the complexity is the bidirectionality of the relationship between environmental factors and disease. While environmental factors can lead to telomere length change, which in turn impacts disease risks, the disease itself and its progression and treatment may lead to telomere length change as well. Therefore, estimating disease risks from phenotypically measured telomere length at any given time point is challenging and imprecise without fully accounting for the myriad potential confounding factors.

Finally, our analysis reflects our best efforts to accurately estimate the incidences, DALYs, and excesses in both that may be attributed to just one source of telomere length variation—genetically predicted telomere length. Data were pulled from multiple sources, including a meta-analysis and multiple population health databases. While every effort was made to maximize consistency across the study populations (see S1 File), the values presented here are subject to change as new methods and research provide more accurate epidemiological data. Nevertheless, our analysis of the population disease burden due to genetically determined telomere length provides the first such estimation on a population level without confounders from environmental exposure, lifestyle factors, and diseases. Future studies using telomere length as a biomarker for disease and risks need to carefully consider the separate effects of genetic, environment (both prenatal and postnatal) and lifestyle factors, and their potential interactions.

## Supporting information

**S1 File.**
(PDF)

## Acknowledgments

We thank Dr. Elizabeth Blackburn and Ms. Dana Smith for insightful discussions and critical reading of the manuscript.

## Author Contributions

**Conceptualization:** Ekaterina Protsenko, David Rehkopf, Aric A. Prather, Elissa Epel, Jue Lin.

**Data curation:** Ekaterina Protsenko, Jue Lin.

**Formal analysis:** Ekaterina Protsenko, Jue Lin.

**Investigation:** Ekaterina Protsenko, Elissa Epel, Jue Lin.

**Methodology:** Ekaterina Protsenko, David Rehkopf, Jue Lin.

**Supervision:** Elissa Epel, Jue Lin.

**Validation:** Jue Lin.

**Visualization:** Ekaterina Protsenko, Jue Lin.

**Writing – original draft:** Ekaterina Protsenko, Jue Lin.

**Writing – review & editing:** Ekaterina Protsenko, David Rehkopf, Aric A. Prather, Elissa Epel, Jue Lin.

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
