## [Decision Letter · Decision Letter 0]

14 Aug 2020

PONE-D-20-19301

Are Long Telomeres Better Than Short?: Relative Contributions of Genetically Predicted Telomere Length to Neoplastic and Non-Neoplastic Disease Risk and Population Health Burden

PLOS ONE

Dear Dr. Lin,

Thank you for submitting your manuscript to PLOS ONE. After careful consideration, we feel that it has merit but does not fully meet PLOS ONE’s publication criteria as it currently stands. Therefore, we invite you to submit a revised version of the manuscript that addresses the points raised during the review process.

Please follow the suggestions of the reviewer for your revision.

We look forward to receiving your revised manuscript.

Kind regards,

Gabriele Saretzki, PhD

Academic Editor

PLOS ONE

Journal Requirements:

2.Thank you for stating the following in the Competing Interests section:

[I have read the journal's policy and the authors of this manuscript have the following competing interests: Jue Lin is a co-founder and minor stock holder of Telomere Diagnostics Inc. ].

3. We note you have included a table to which you do not refer in the text of your manuscript. Please ensure that you refer to Table 4 in your text; if accepted, production will need this reference to link the reader to the Table.

Additional Editor Comments (if provided):

Please follow the comments of the reviewer for your revision.

Reviewers' comments:

Reviewer's Responses to Questions

**Comments to the Author**

1. Is the manuscript technically sound, and do the data support the conclusions?

Reviewer #1: Yes

2. Has the statistical analysis been performed appropriately and rigorously? 

Reviewer #1: I Don't Know

3. Have the authors made all data underlying the findings in their manuscript fully available?

Reviewer #1: Yes

4. Is the manuscript presented in an intelligible fashion and written in standard English?

Reviewer #1: Yes

5. Review Comments to the Author

Reviewer #1: I uploaded my review comments in a separate word document, please see that for my specific comments on this paper . Many Thanks for the opportunity to review this.

6. PLOS authors have the option to publish the peer review history of their article (what does this mean?). If published, this will include your full peer review and any attached files.

Reviewer #1: **Yes: **Abraham Aviv

---

## [Author Response · Author response to Decision Letter 0]

16 Sep 2020

Dear Drs. Saretzki and Aviv, 

We thank you for your insightful and constructive comments on our manuscript titled “Are Long Telomeres Better Than Short?: Relative Contributions of Genetically Predicted Telomere Length to Neoplastic and Non-Neoplastic Disease Risk and Population Health Burden”. We have carefully reviewed all the comments and addressed each of them either by revising the manuscript accordingly or providing explanations and clarifications. Please see attached the revised manuscript and the response letter. Thank you again for the opportunity to submit our manuscript to your journal and we look forward to hearing back from you.

Sincerely,

Jue Lin, Ph. D.

Department of Biochemistry and Biophysics

Room S316, 600 16th Street

San Francisco, CA 94158

Phone (415) 476-7284

Email: jue.lin@ucsf.edu

Response to reviewer’s comments

Introduction

-L81-84. “These estimates from family-based studies…”: While I agree with the statement, robust associations of leukocyte TL (LTL) were observed between newborns and their parents. One might argue that a shared intrauterine environment contributes to these associations. Still, the impact of the environment on LTL, in my view, is much less than that of the genetic makeup of the individual.

Authors’ response: We have revised this sentence to reflect the contribution of a shared intrauterine environment. The new version now reads as follows:

“These estimates from family-based studies reflect both the genetic contribution and shared environmental factors due to the relatedness in the study participants, as well as a shared intrauterine environment in some cases, and therefore are likely to be higher than the true portion of telomere length variation determined by genetic inheritance”. 

-L 87-89. “A recent study using genome-wide complex trait analysis…”: Statement is based on a study that used salivary DNA TL measurements by qPCR. Findings are less robust than those based on larger datasets generated using LTL measured by either Southern blotting or qPCR.

Authors’ response: We agree with Dr. Aviv that TL results using salivary DNA are more likely impacted by preanalytical factors, therefore less robust. We have added the following sentence in the revised version. 

“However, it should be noted that this study used salivary DNA for TL measurements, which are less robust than DNA from leukocytes.”

-L 108-115. “It has been proposed that the opposing directions…”: Statements are imprecise and should be rewritten. Ref 53 discusses the role of evolutionary forces, which by definition work principally during the reproductive years, in fashioning an optimal TL in humans. It underscores that in contemporary humans, the lasting impact of these forces is expressed during the post-reproductive years. The selective evolutionary forces on TL is also implicit in the narrative of ref 54 and PMID 26936823. Moreover, other publications further elaborate on this idea (PMID 30631124, PMID: 32427393).

Authors’ response: We thank Dr. Aviv for this very helpful suggestion. We have revised the statement and it now reads as follows:

“It has been proposed that the opposing directions of these associations imply that human telomere length has evolved to balance the disease risks imposed by both short and long telomeres[53, 54], and thereby achieving an optimal length over successive generations. It would seem that this argument fails to consider the fact that the manifestation of the majority of the diseases associated with long or short telomeres, with the exception of a number of rare cancers (e. g. neuroblastoma and testicular cancer, some types of leukemia and lymphoma and type I diabetes mellitus), happens in later life, well past the reproductive period to have real evolutional pressure on the population. However, this finding among contemporary humans could alternatively suggest that selection forces have acted on telomere length precisely to lessen the impact of degenerative and neoplastic diseases during the reproductive years [55, 56].”

Discussion

-L239-242. When we apply our method to estimate the excess incidence of these conditions, we find that…”: Indeed, this is the key point of the paper, yet it is not emphasized in the abstract.

Authors’ response: We revised the abstract to emphasize this point. The revised Results section of the abstract is shown below:

“Results: Our analysis shows that, despite the markedly larger ORs of neoplastic disease, the large incidence of degenerative diseases causes the excess incidence attributable to gTL to balance that of neoplastic diseases.”

-L347-349. “Second, as environmental factors can influence telomere length throughout the whole lifespan…”: No doubt, this statement is true, particularly during infancy/early childhood, when TL shortening is rapid, but the overall magnitude of the effect is minor compared to the inter-individual TL variation at birth. This is barely appreciated from studies that generate T/S data by qPCR.

Authors’ response: We agree with Dr. Aviv that the magnitude of TL change over adult life is much smaller compared to the inter-person variation at birth. We have added a sentence to reflect this (see below).

“Second, as environmental factors can influence telomere length throughout the whole lifespan, and perhaps differentially during different developmental stages (childhood, adulthood and geriatric), the disease risks caused by short or long telomere length can change accordingly. Although it should be noted that compared to the inter-individual TL variation at birth, the overall magnitude of the effect of environmental factors is smaller [69].”

---

## [Editor Report · Decision Letter 1]

22 Sep 2020

Are Long Telomeres Better Than Short?: Relative Contributions of Genetically Predicted Telomere Length to Neoplastic and Non-Neoplastic Disease Risk and Population Health Burden

PONE-D-20-19301R1

Dear Dr. Lin,

We’re pleased to inform you that your manuscript has been judged scientifically suitable for publication and will be formally accepted for publication once it meets all outstanding technical requirements.

Kind regards,

Gabriele Saretzki, PhD

Academic Editor

PLOS ONE

Additional Editor Comments (optional):

The authors corresponded satisfactorily to the reviewers comments and improved the manuscript. Please add "types"or "entities" or something similar to line 47 in the abstract since the sentence"..from the 9 cancer" seems incomplete.
---

## [Editor Report · Acceptance letter]

28 Sep 2020

PONE-D-20-19301R1 

Are Long Telomeres Better Than Short?: Relative Contributions of Genetically Predicted Telomere Length to Neoplastic and Non-Neoplastic Disease Risk and Population Health Burden 

Dear Dr. Lin:

I'm pleased to inform you that your manuscript has been deemed suitable for publication in PLOS ONE. Congratulations! Your manuscript is now with our production department. 

Kind regards, 

on behalf of

Dr. Gabriele Saretzki 

Academic Editor

PLOS ONE